# A Method for Preparing Surface Sub-Microstructures on Sapphire Surfaces Using Femtosecond Laser Processing Technology

Kaixuan Wang, Jun Chen, Yubin Zhang [ID], Qingzhi Li, Feng Tang [ID], Xin Ye *[ID] and Wanguo Zheng *

Research Centre of Laser Fusion, China Academy of Engineering Physics, Mianyang 621900, China; wangkaixuan20@gscaep.ac.cn (K.W.); chenjun19950110@163.com (J.C.); zhangyub@mail.ustc.edu.cn (Y.Z.); dearlqz@163.com (Q.L.); tangfengf3@caep.cn (F.T.)
* Correspondence: xyecaep@mail.ustc.edu.cn (X.Y.); wgzheng_caep@sina.com (W.Z.)

**Abstract:** Femtosecond laser processing technology is an advanced sub-micro-processing technique that enables the non-contact processing of various materials. This technology can be used to apply sub-micro structures for purposes such as hydrophilicity enhancement, optical transmittance improvement, and photonics detection. However, when it comes to processing micro/nanostructures on highly brittle materials using femtosecond lasers, there are challenges such as low processing efficiency, generation of debris, and microcracking. In this paper, we propose a method called the out-of-focus femtosecond laser direct writing technique combined with wet etching. This method offers simplicity, speed, and flexibility in preparing dense, large-area sub-microstructured surfaces on the brittle material sapphire. Our detailed investigation focuses on the impact of laser processing parameters (direct writing period, distance of focusing, direct writing speed, etc.) on the sub-microstructures of $Al_2O_3$ surfaces. The results demonstrate that this method successfully creates embedded sub-microstructures on the sapphire surface. The microholes, with a diameter of approximately 2.0 μm, contain sub-micro structures with a minimum width of $250 \pm 20$ nm. Additionally, we conducted experiments to assess the optical transmittance of sapphire nanostructures in the range of 350–1200 nm, which exhibited an average transmittance of approximately 77.0%. The water contact angle (CA) test yielded a result of $52 \pm 2°$, indicating an enhancement in the hydrophilicity of the sapphire nanostructures with only a slight reduction in optical transmittance. Our efficient fabrication of sub-microstructures on the sapphire surface of highly brittle materials offers a promising method for the production and application of brittle materials in the field of micro-optics.

**Keywords:** femtosecond laser processing; sapphire; surface sub-microstructures; SERS



## 1. Introduction

Advanced manufacturing technology is the basis for scientific and industrial progress [1–4]. Femtosecond laser processing technology has become an effective tool for the preparation of micro and nanostructures because of its high flexibility, high reproducibility, fast processing speed, high efficiency, and high accuracy [5–7]. Femtosecond laser processing technology allows maskless non-contact one-step processing in atmospheric, solution environments, with the ability to prepare sub-microstructures with high precision and quality on virtually any material surface [8,9]. Femtosecond laser processing can prepare a variety of sub-wavelength structures on material surfaces, including cones, micron-nano spikes, holes, and pits [10–12]. Patterned material surfaces can significantly improve the physical, chemical, and biological properties of the target material [13–15]. Conventional methods to fabricate surface sub-microstructures include electron beam lithography, photolithography, plasma etching, wet etching, and nanoimprinting [16–18]. All of these techniques are limited by multiple aspects, such as process complexity, materials, substrate shape and size, and also high cost and multi-stage processes. Femtosecond lasers have been widely studied for

their ability to induce surface sub-microstructures on metal, dielectric and semiconductor surfaces [19–21]. The surface sub-microstructures of materials are closely related to the laser injection energy, laser wavelength, number of pulses, and beam polarization.

Sapphire is widely used in aerospace, semiconductors, photonics, and other fields due to its excellent broad spectral transmittance, mechanical strength, thermal stability, electrical properties, and chemical stability [22–24]. Sapphire is an excellent optical material because of its ultra-high hardness and chemical resistance, so traditional mechanical processing and chemical wet etching at the micro- and nanoscale are no longer applicable. In recent years, sub-microstructures prepared by femtosecond laser processing technology has gained more and more widespread attention in the fields of micro-optics, microlens arrays, microfluidic devices, and optical detection [25–27]. Because the femtosecond laser has an ultra-short pulse ($10^{-15}$ s) and high peak power (power density after focusing up to $10^{22}$ w/cm$^2$) femtosecond laser processing technology can be used to prepare sub-microstructures on the surface of hard and brittle materials such as sapphire [28]. However, the processing of hard materials usually needs to be ablated with high-energy density lasers, which does not allow for high precision and efficient processing [29]. In addition, femtosecond laser interaction with sapphire is a nonlinear absorption generating Coulomb micro-explosion process. The surface of the prepared sapphire sub-microstructures shows significant roughness and micro-cracks [30]. This makes it difficult to meet the requirement of high surface smoothness for optical devices. Liu's group proposed a femtosecond laser depth scribing technique to fabricate high-aspect-ratio bionic microstructures by adding a silica sacrificial layer on sapphire, resulting in microstructures with a maximum aspect ratio of 80:1 [31]. Wen's group deposited a gold nanofilm on sapphire, which allowed the laser energy to be deposited uniformly on the sapphire surface, thus increasing the material removal rate of sapphire by ~2.24 times [32]. Lu's group proposed vector scanning subtractive manufacturing to fabricate micro-optical components. It enhanced the processing efficiency of sapphire by 100 times and effectively reduced the stress and reduced the roughness of the material surface to 28 nm [33]. All these efforts were important in the development of micro-optics for hard materials. However, achieving the preparation of hard materials with high efficiency in manufacturing at the nanoscale remains a challenge. In this paper, a combination of out-of-focus femtosecond laser processing and wet etching was used for the rapid fabrication of random surface sub-microstructures on the hard material sapphire. In addition, out-of-focus femtosecond laser fabrication reduced the accumulation of laser energy and lowered internal stresses in the sample, reducing surface cracking compared to conventional femtosecond laser technology fabrication techniques. HF wet etching was used to solve the problem of excessive roughness and to improve the efficiency of processing [34].

The remarkable feature of the prepared surface sub-microstructures was the fast, efficient, and randomized nanostructure period much smaller than the laser wavelength. During femtosecond laser processing, high energy femtosecond lasers produce large amounts of debris when applied to the sapphire surface. The debris and particles scatter and block the femtosecond laser, resulting in energy attenuation that hinders subsequent surface processing [35]. The debris also attached to the surface of the sample, resulting in a significant roughness on the surface of the sample, affecting its optical properties. Therefore, for high-quality sapphire surface sub-microstructures, we solved the problem of fragility and fragmentation with out-of-focus femtosecond laser processing of the target material and wet etching techniques. In this paper, the relationship between the parameters of the femtosecond laser (direct writing speed, scanning period, distance of focusing, etc.) and the etching of the HF with the sapphire sub-microstructures were investigated. Excellent sub-microstructures of the sapphire surface were obtained (the diameter of the microholes was ~2.0 μm, and the period of the nanostructures in the microhole was ~400 nm). These sub-microstructures with special morphology and unique properties attracted our keen attention. Moreover, the experimentally measured sub-microstructure of the sapphire surface was a super-wetting functional surface with a contact angle of ~54 ± 2° (flat sapphire CA

~74 $\pm$ 2°). At the same time, its transmittance in the visible-near-infrared was only slightly reduced while optical transmittance was increased in the mid-wave infrared band. The current main investigation on femtosecond laser–sapphire interaction is the mechanism of sub-microstructure formation. For example, Miyagawa's group deposited different materials on sapphire samples to control the laser-induced periodic surface structure and obtained low-period nanostructures [36]. However, there are relatively few studies on the application of sapphire submicron structures. In this paper, by investigating the nature of sapphire submicron structures, it is found that there are relatively few studies on sapphire submicron structures for SERS probing. In this paper, femtosecond laser processing combined with wet etching technique is proposed to establish a SERS detection method with low cost, excellent stability, and reproducibility. This provides technical support for the application of the sapphire submicron structure.

## 2. Experimental Section

$Al_2O_3$ (crystal orientation was c-plane, thickness was ~0.5 mm) wafers were first cleaned with acetone in an ultrasonic bath, and then the $Al_2O_3$ was processed with a femtosecond laser. Femtosecond lasers (the laser beam was linearly polarized with a central wavelength of 800 nm, a pulse width of 120 fs, and frequency of 1 kHz) relied on a coaxial charge-coupled device (CCD) and a lens aggregation system, where the laser can be focused quickly and accurately on the material surface. The focusing objective was $20\times$, NA. = 0.40 and the beam spot was ~2.4 μm (the beam spot was calculated as follows: D = 1.22λ/NA., λ was the working wavelength, and NA. was the numerical aperture of the objective lens). The intensity distribution of the laser followed a Gaussian distribution. Femtosecond laser processing depended on a mechanical platform to move the sample (positioning accuracy of 40 nm, 40 nm, and 100 nm in X, Y, and Z directions, respectively). The experiments were carried out with an out-of-focus femtosecond laser raster-type scanning with a scanning speed of 3500~11,500 μm/s and a scanning pitch of 2 μm, The laser fluence was 25 J/cm². Large-area sapphire sub-microstructures (4 mm $\times$ 4 mm) were successfully prepared in 30 min. The debris generated during femtosecond laser processing could be removed cleanly by etching with a 20% wt. HF solution for 2.0 h. The morphology of the nanostructures was observed using scanning electron microscopy (SEM). The standard type contact angle meter tester was used to measure the hydrophilic nature of the structures using 30 μL of distilled water. Transmittance of the sapphire sub-microstructures was measured by UV–visible near-infrared spectrophotometer and Fourier-transform infrared spectrometer.

During direct writing of the femtosecond laser on the $Al_2O_3$ surface, the ablation of the femtosecond laser produced a high-pressure environment causing damage to the sapphire and debris to be ejected and reflected to the sample surface. Sputtering of debris on the sapphire surface caused some absorption and masking of the laser energy which disrupted the formation of the sub-microstructure. If only a femtosecond laser was used to ablate the sapphire surface it would have a significant roughness, as shown in the SEM image of Figure 1A. This was because of the sapphire debris adhering to the surface of the sub-microstructures. The debris on the surface of the sapphire sub-microstructures was cleaned and the nanostructures of the microholes were visible, as shown in the SEM image of Figure 1B. When the cleaning time was 1 h, the nanostructures within the microholes were not fully exposed, as shown in Figure S1. The reason for choosing HF etching was due to the transformation of the irradiated sapphire from the crystalline to the amorphous/polycrystalline phase by the femtosecond laser [37]. The amorphous/polycrystalline region was rapidly etched away by the HF solution, and the etching rate of sapphire in the amorphous region was much higher than that in the crystalline region, with etch selectivity exceeding $10^4$ [38].

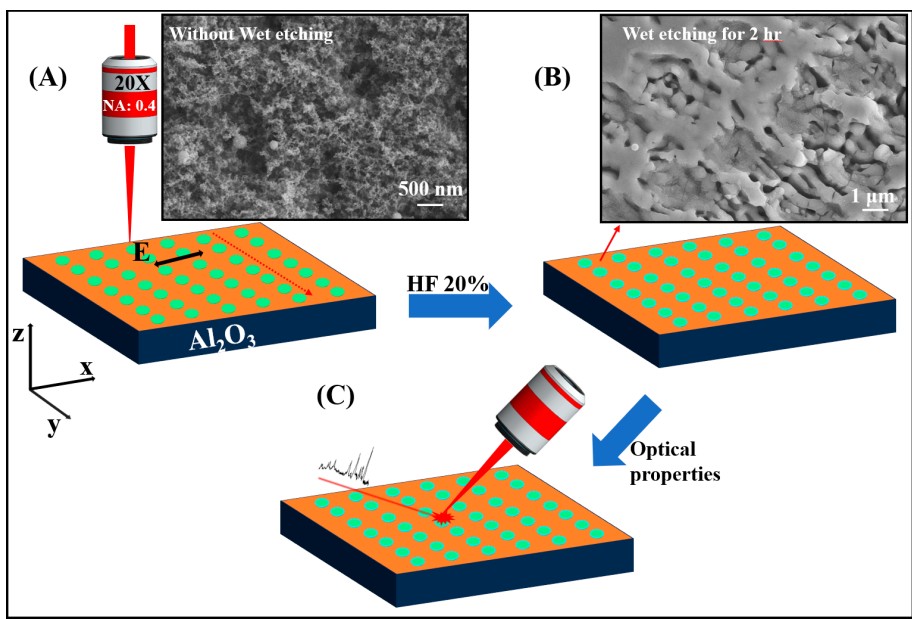

**Figure 1.** Schematic of Al$_2$O$_3$ sub-microstructure processing: (**A**) femtosecond laser direct writing with 0 h etching time; (**B**) HF etching with 2 h etching time; (**C**) optical properties test.

## 3. Results and Discussion

### 3.1. Effect of the Distance Out-of-Focus of the Femtosecond Laser Beam Spot on the Sub-Microstructures

High-energy laser pulses had a strong non-linear effect on the sapphire material resulting in low structural accuracy. Generally, the damage threshold on the surface of the sapphire was lower than the damage threshold on the interior of the sapphire. When the focus of the femtosecond laser beam spot was above the surface of the sample to process the material, it was able to regulate the energy and effectively reduce the heat-affected zone and microcracks on the surface of the material. In this section, the effect of the distance between the focal point of the laser beam spot and the sample surface on the sub-microstructure of the sapphire surface was investigated. When the femtosecond laser focuses on the sapphire surface through the objective lens, part of the beam spot was inside the sapphire. By adjusting the three-dimensional displacement table, the beam spot of the femtosecond laser was made deep or far away from the surface of the sample. As shown in the inset of Figure 2, the positions of the beam spots were schematically shown as −6 um, −3 um, 0 um, +3 um, +6 um, and +9 um in that order ("0" indicates that the beam spot focus was on the surface of the sample, "−" indicates that the beam spot focus was inside the surface of the sample, "+" indicates that the beam spot focus was above the sample surface). It is shown in Figure 2A,B that when the laser was focused on the interior of the sapphire at −6 μm and −3 μm, only a few microholes (microholes diameter was 2.0 μm) and some cracks were obtained. These microholes were caused by micro-explosions, thermal melting, and shock waves generated when the femtosecond laser interacts with the sapphire. Because the femtosecond laser is focused inside the sapphire material, it only takes a small average power to induce micro-explosions that generate extreme high temperatures and pressures [39]. The high temperatures and pressures generated in a very small space create powerful shock waves that can cause the material to rapidly spread around to form microholes and microcracks.

Figure 2C,D show that when the beam spot focus was 0, +3 μm above the sample surface, the experiment successfully prepared a large number of uniform sub-microstructures on the sapphire surface (the minimum width of the submicron structure was 250 ± 20 nm, as shown in Figure S2). There was a portion of random sub-microstructures inside the microholes, and another portion was laser-induced periodic structures (LIPSS) (the period was 400 nm, λ/n < P < λ/2n, n = 1.70) [40,41]. The nanostructure inside the microholes was

parallel to the direction of the electric field vector. The generation of LIPSS was influenced by the energy of the femtosecond laser and the spatial overlap rate of the beam spot. Therefore, LIPSS could only be obtained under certain laser parameter conditions, and it would not form under excessive or insufficient laser ablation conditions. When the laser focus was far away from the sapphire surface, +6 μm, there were fewer nanostructures isolated far between each microhole. When the distance increased to +9 μm, the distance between pores became more distant, and there were almost no nanostructures inside the microholes, as shown in Figure 2E,F. The main reason was that when the focus of the femtosecond laser was far away from the sapphire surface, the laser intensity gradually approached the damage threshold of the sapphire surface. Due to the low energy, it was unable to form a large number of sub-microstructures, only causing micro-damage to the sapphire surface. When the femtosecond laser was focused inside the sapphire, internal damage and surface defects could not form nanostructures on the sapphire. When the focus of femtosecond laser was close to the surface of the sample at 0–3 μm, surface sub-microstructures (random nanostructures and some LIPSS) were successfully prepared. The reason for the formation of LIPSS was that when femtosecond laser pulses continuously irradiated the surface of sapphire, transient surface plasmon states appeared on the sapphire surface, which interfered with subsequent laser radiation at the solid–gas interface [42]. It generated periodic energy ripples. The portion of interference with high laser energy exceeding the sapphire's ablation threshold caused the sapphire to transition from crystalline to amorphous. The low laser energy part in the interference remained almost unchanged. Finally, LIPSS was formed after hydrofluoric acid corrosion. We chose a femtosecond laser out-of-focus distance in the range of 0–3 μm. This kind of process tolerance could effectively perform repetitive experiments.

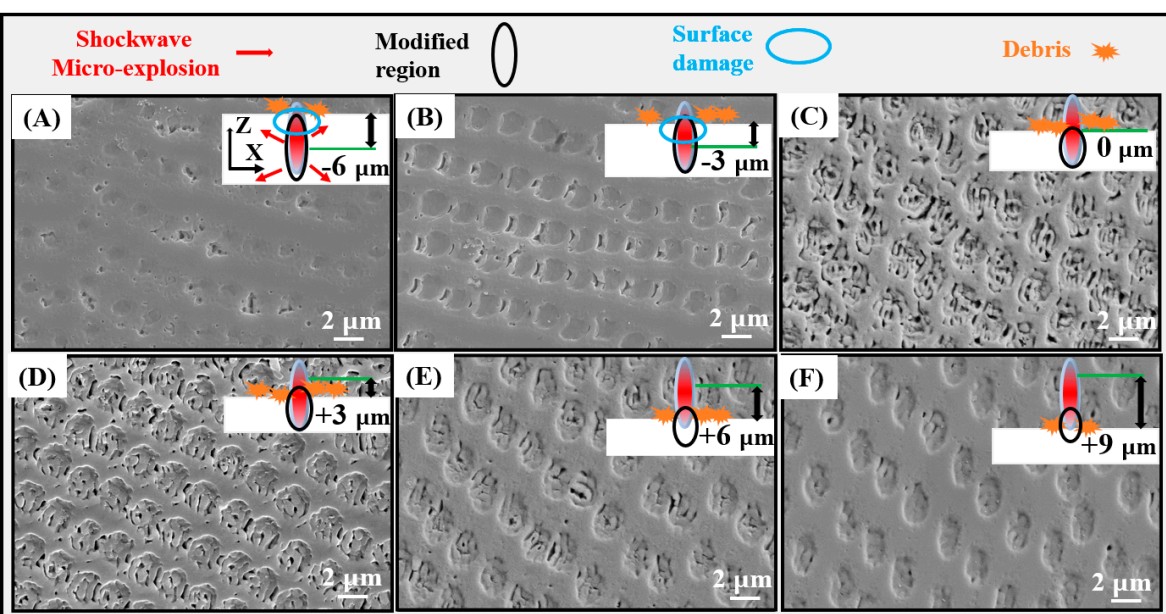

**Figure 2.** Femtosecond laser beam spot with different out-of-focus distances: (**A**) −6 μm; (**B**) −3 μm; (**C**) 0 μm; (**D**) +3 μm; (**E**) +6 μm; (**F**) +9 μm ("−" means the focus of the beam spot was inside of the sample, "+" means the focus of the beam spot was above the sample).

### 3.2. Effects of Scanning Speed and Space on Sub-Microstructures

As is well known, the formation of sub-microstructures on material surfaces was closely related to many laser parameters such as scanning speed, scanning space, laser power, etc. In this section, the formation of sub-microstructured surfaces of sapphire was explored in terms of the femtosecond laser direct writing speed and scanning space. Firstly, we analyzed the morphology of the sub-microstructured surface of sapphire at different femtosecond laser direct writing speeds. Figure 3A–H shows the SEM images

of the sample under different laser scanning speeds, which are 500 μm/s, 1500 μm/s, 3500 μm/s, 4500 μm/s, 5500 μm/s, 7500 μm/s, 9500 μm/s, and 11500 μm/s, respectively. In addition, the parameters used for the femtosecond laser direct writing in this paper were as follows: the objective was 20×, NA. = 0.4, laser fluence was 25 J/cm$^2$, beam spot focus was at +3 μm above the sapphire surface, and 20% HF treatment for 2 h. When the direct writing speed was 1500 μm/s, microholes (Region I) and irregular dense nanostructures (Region II) were formed on the sapphire surface, as in Figure 3B. This was due to the high number of pulses and high beam spot overlap of the femtosecond laser resulting in an excessively high accumulation of energy on the sapphire surface. This created a partially disordered and dense sub-microstructure. As the laser direct writing speed decreased to 500 μm/s, the overlap of the femtosecond beam spot was even higher to excite more intense surface equipartitioned excitations, as in Figure 3A. The structure of Region I in Figure 3B transforms into the structure in Figure 3A. Similarly, when the direct writing speed was increased to 3500 μm/s–5500 μm/s, we successfully prepared regular dense sapphire sub-microstructures, which formed microholes and nanostructures embedded within the microholes. The structure of Region II in Figure 3B was transformed into the structures in Figure 3C–E. The number of laser pulses and beam spot overlap rate were reduced when the laser direct writing speed was greater than 7500 μm/s. The sapphire surface only formed micropores (the diameter was ~2.0 μm), and it was difficult to form dense nanostructures within the microholes, as shown in Figure 3F–H. In Figure 3, there were some irregularly arranged microholes, and some microholes were connected. This was due to the oscillating effect of the high-speed 3D displacement stage during the laser writing process. The experiments considered the efficiency and homogeneity of the preparation of sub-microstructures on the sapphire surface. The laser direct writing speed was experimentally determined to be v = 3500 μm/s. We also found that the diameter of the experimentally prepared microhole structure was ~2.0 μm smaller than the theoretically estimated femtosecond beam spot size (~2.44 μm). The reason is that the preparation method we use leads to the creation of smaller microholes.

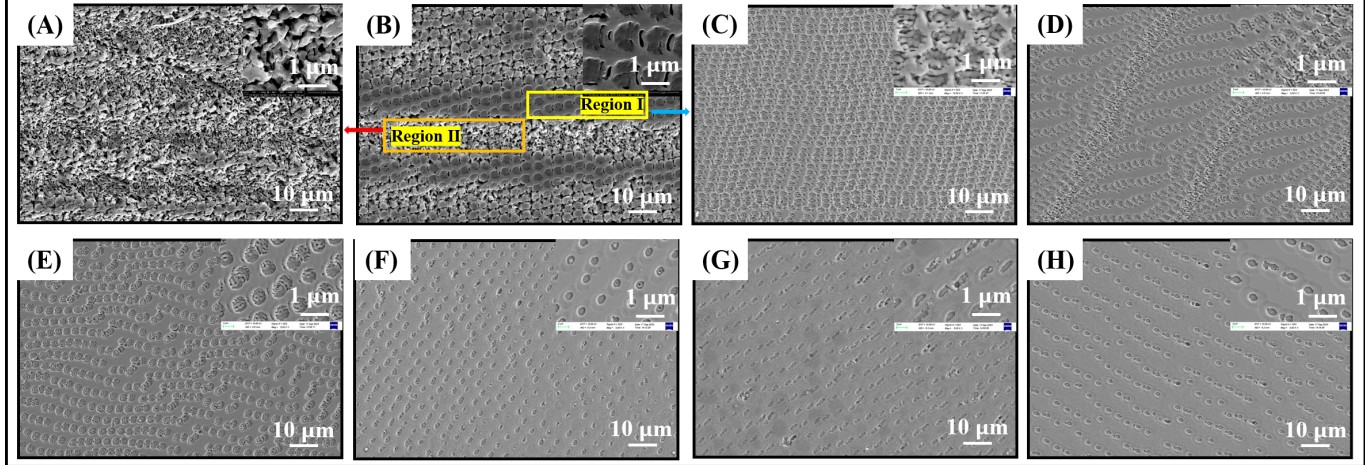

**Figure 3.** SEM images of different femtosecond laser direct writing speeds: (**A**) 500 μm/s; (**B**) 1500 μm/s; (**C**) 3500 μm/s; (**D**) 4500 μm/s; (**E**) 5500 μm/s; (**F**) 7500 μm/s; (**G**) 9500 μm/s; (**H**) 11,500 μm/s.

Secondly, the experiment further explored the effect of scanning space on the sub-microstructure. The parameters of the femtosecond laser direct writing were as follows: the direct writing speed was 3500 μm/s, the objective was 20×, NA. = 0.4, laser fluence was 25 J/cm$^2$, and the out-of-focus distance was +3 μm. As shown in Figure 4A, when the scanning space was 2.0 μm, microholes appeared on the surface of the sapphire sub-microstructures interconnected with the microholes. The nanostructures (period was 400 nm) in the microholes were also interconnected. This was because the diameter of the femtosecond beam spot was slightly larger than the direct writing spacing to cause the

microholes to interconnect. As shown in Figure 4B, when the spacing was 3.0 μm, the spacing was larger than that of the femtosecond beam spot diameter, but there were still some microholes connections due to the jitter factor of the three-dimensional displacement stage during the direct writing processing. When the write spacing was further increased to 5.0 μm, the increase in spacing prevented the laser beam spots from overlapping and eliminated the effects of the 3D stage oscillations, as shown in Figure 4C. From Figure 4, we found that when the scanning space was 2.0 μm, the femtosecond beam spot almost overlapped in the direct writing process, and it was easier to obtain random sub-microstructures. Among them, microholes were also connected with microholes in a large range, and the microholes also contained dense nanostructures. This provided a good platform for the enrichment of the SERS substrate with the substances to be detected.

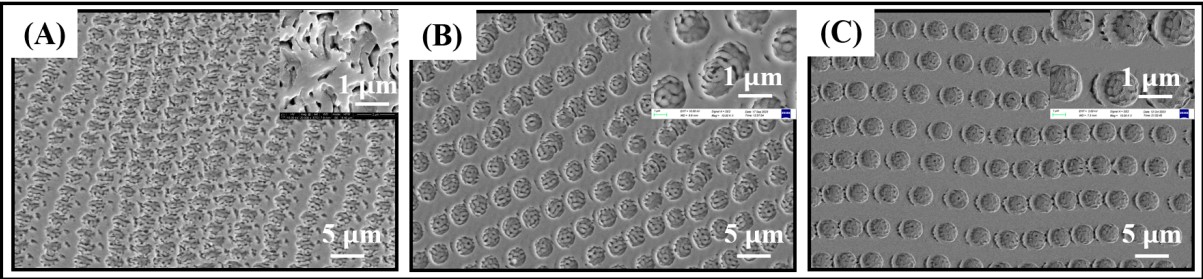

**Figure 4.** Pitch of different femtosecond laser direct writing: (**A**) 2.0 μm; (**B**) 3.0 μm; (**C**) 5.0 μm.

### 3.3. The Wettability and Optical Properties

The wetting properties of the sapphire micro- and nanostructured surfaces were measured. The contact angle (CA) was $76 \pm 2°$ as measured on the flat sapphire surface, as shown in Figure 5A. The contact angle of the sample of Figure 5B was $79 \pm 2°$ treated using femtosecond laser ablation only. Its contact angle was slightly increased indicating an increase in the surface roughness of the sapphire. Because it was not wet etched, there was a lot of debris on the surface of the sapphire sub-microstructures. Figure 5C–E showed the contact angles of the samples after femtosecond laser direct writing and HF acid etching, which were $52 \pm 2°$, $63 \pm 2°$, and $66 \pm 2°$, respectively (corresponding to Figure 4A–C, respectively). The hydrophilicity was enhanced because the roughness of the sub-microstructures on the sapphire surface was reduced after wet etching. Moreover, the molecular bonding of sapphire was Al-O which was easier to combine with water to become more hydrophilic. When the scanning space of the direct writing was 2 μm, it connected a large number of micropores, and the nanowires in the microholes were also interlaced with each other, which made the surface area larger and more hydrophilic. And when the scanning space gradually increased, the connectivity between the microholes became less and the contact surface also decreased; then, they became less hydrophilic. Therefore, the sub-microstructures on the sapphire surface after femtosecond laser direct writing and wet etching showed high uniformity and low roughness, as shown in Figure S3.

We tested the optical properties of the sub-microstructures on the sapphire surface. As shown in Figure 6A, the average transmittance of the sapphire sub-microstructures without wet etching ~65.0% was less than that of the flat sapphire ~85.0%, in the range of 350–1200 nm. This was because the roughness of the sapphire surface after direct laser writing caused light scattering and absorption. The roughness of the surface of the sapphire sub-microstructures was reduced after the wet treatment, and its average transmittance increased to ~77.0%. However, it was ~8.0% lower than the transmittance of flat sapphire. In the 2.5–6.5 μm band range, the average transmittance of the wet-etched sapphire sub-microstructures of ~88.2% was ~2.2% higher than that of the flat sapphire of ~86.0%, while the average transmittance of the sapphire micro- and nanostructures without wet etching was ~80.0%, as shown in Figure 6B. According to the effective medium theory (EMT), the period of the subwavelength structure is determined by the following equation: $p/\lambda = (n_2 + n_1 Sin\theta_{max})^{-1}$, p is the anti-reflective periodic subwavelength structure,

$n_2 = 1.77$ is the refractive index of sapphire, $n_1 = 1$ is the refractive index of air, and $\theta_{max} = 90°$ is the maximum of the incident angle. The periodicity of the subwavelength transmittance-enhancing structure determines the working wavelength, so when the infrared wavelength $\lambda$ is 3 μm, the period of the transmittance-enhancing subwavelength structure should be less than 1.69 μm, and the experimentally prepared sapphire submicrometer has a period of 400 nm. Therefore, it improves the infrared transmittance [43]. And the height and shape of the pattern determine the transmittance enhancement performance, and the optimal height of the structure when the reflectivity is minimum is $h_{min} = \lambda/4(n_1n_2)^{-1}$, where the infrared wavelength $\lambda$ is 3 μm, and $n_1$ and $n_2$ are the refractive indices of the sapphire and air, respectively. Therefore, the height of the permeation-enhancing subwavelength structure should be greater than 0.56 μm. The depth of the sapphire sub-microscopic structure tested in our experiments is greater than 580 nm, as shown in Figure S4. Similarly, the experimentally prepared sub-microscopic structures also enhance the IR transmittance. The transmittance of the sub-microstructures on the sapphire surface without wet-assisted etching was significantly lower than that of flat sapphire, as shown in Figure 6A,B. Therefore, HF wet etching after laser direct writing was necessary because it could enhance the transmittance of the sub-microstructures on the sapphire surface.

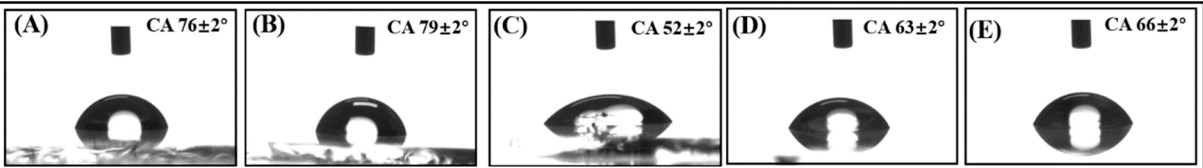

**Figure 5.** Contact angles of different sapphire sub-microstructures: (**A**) flat sapphire; (**B**) femtosecond laser direct writing without HF cleaning; contact angles of different femtosecond laser direct writing space with HF cleaning: (**C**) 2 μm; (**D**) 3 μm; (**E**) 5 μm.

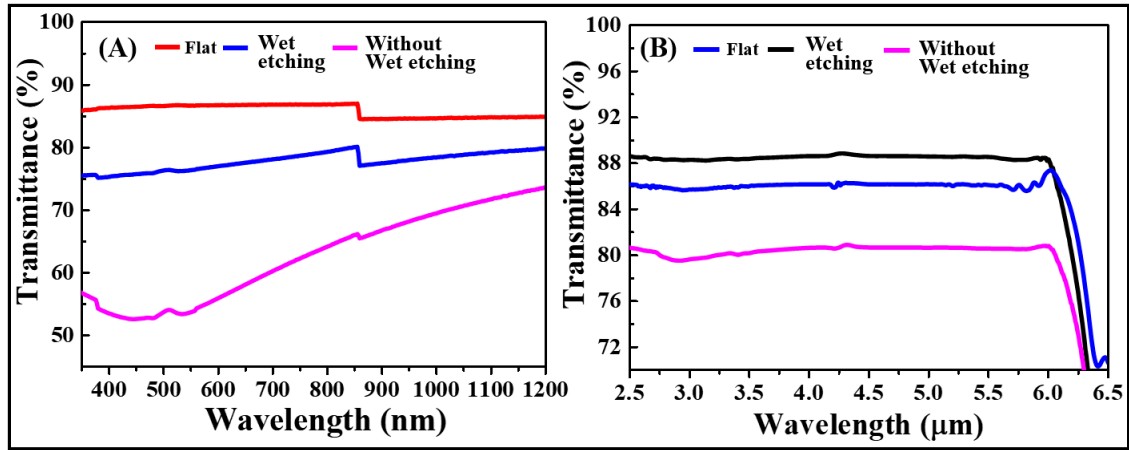

**Figure 6.** Transmittance of sub-microstructured sapphire: (**A**) transmittance in the range of 350–1200 nm; (**B**) transmittance in the range of 2.5 μm–6.5 μm.

## 4. Conclusions

In conclusion, sub-microstructures were successfully prepared on the surface of sapphire, a high-hardness brittle material, using the technique of out-of-focus femtosecond laser direct writing combined with HF wet etching. This was a low-cost, rapid, flexible, and efficient method to prepare random $Al_2O_3$ surface sub-microstructures. We could flexibly adjust the size of the area density of the sapphire microstructures by adjusting the femtosecond laser's direct writing speed, direct writing space, and out-of-focus distance, and reduce the surface roughness of the sub-microstructures by using wet etching. The speed of the femtosecond laser was 3500 μm/s, the direct writing space was 2.0 μm, the

out-of-focus distance was at 0–3 μm, and using HF etching of 2.0 h it was possible to prepare large-area microholes (2.0 μm), nanogrids (~250 nm ± 20 nm) (nanostructures in microholes) on the surface of the sapphire. The experiments showed that the average transmittance of the sapphire sub-microstructure was ~77.0% without much reduction in the range of 350–1200 nm, and the average transmittance of ~88.2% in the range of 2.5–6.5 μm band achieved increased transmittance. And there was also a very good hydrophilicity where CA was 52.0 ± 2°. Random, high-density sub-microstructures on sapphire surfaces have promising applications in micro-optics, anti-fog devices, and bio-detection.

**Supplementary Materials:** The following supporting information can be downloaded at: https://www.mdpi.com/article/10.3390/coatings14040481/s1, Figure S1. The sapphire sub-microstructure images with a cleaning time of 1 h; Figure S2. The minimum sub-microstructure within a microhole is 250 ± 20 nm or smaller. Figure S3; The roughness of the sapphire microholes structure is reduced to about 25 nm after hydrofluoric acid cleaning; Figure S4. AFM map of sapphire sub-microstructure.

**Author Contributions:** K.W.: conceptualization, formal analysis, investigation, data curation, writing—original draft, writing—review and editing; J.C.: conceptualization, formal analysis, investigation, data curation; Y.Z.: conceptualization, formal analysis, investigation; F.T.: conceptualization, formal analysis; X.Y.: conceptualization, formal analysis, revision; X.Y. and Q.L.: formal analysis, funding acquisition; W.Z.: conceptualization, formal analysis, revision. All authors have read and agreed to the published version of the manuscript.

**Funding:** This research was funded by the National Natural Science Foundation of China (No. 12304352); The Open Project Program of Key Laboratory for Cross-Scale Micro and Nano Manufacturing, Ministry of Education, Changchun University of Science and Technology (CMNM-KF 202110).

**Institutional Review Board Statement:** Not applicable.

**Informed Consent Statement:** Not applicable.

**Data Availability Statement:** Data are contained within the article and Supplementary Materials.

**Conflicts of Interest:** The authors declare that they have no known competing financial interests or personal relationships that could have appeared to influence the work reported in this paper.

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
