# Peer review of "A Method for Preparing Surface Sub-Microstructures on Sapphire Surfaces Using Femtosecond Laser Processing Technology"

_coatings, doi:10.3390/coatings14040481_

Round 1
Reviewer 1 Report (Previous Reviewer 1)
Comments and Suggestions for Authors
SERS is realized by nanostructures, but the authors are only talking about micro-sized laser spots. When authors take a laser-spoted surface, etch it with HF, and coat it with silver, do you think there are intricately designed nanostructures? This was pointed out to another reviewer and me in the last review, but it's not addressed at all.
And in my last review, I asked for a thorough explanation. I pointed out in my last review the discontinuity at 860 nm in Figure 6a, but the authors only state that this come from the light source switches. If the light source changes, does the material's transmittance change?
This work fails to scientifically validate its claims, and does not follow the advice of the reviewers accordingly. I recommend that this manuscript should be rejected.
Comments on the Quality of English LanguageThe English is okay, but it needs to be modified to fit the engineering style.
Author Response
Please see the attachment.

Reviewer 2 Report (New Reviewer)
Comments and Suggestions for Authors
Authors present method for preparing surface sub-microstructures on sapphire surfaces using femtosecond laser processing technology. Several amendment before considering for publication:-
1- Novelty statement is not clear. Please include in last paragraph in introduction section.
2- Justify the reason depositing 60 nm thick Ag on a sapphire surface sub-microstructured substrate?
3- What is the special properties of Ag? What is advantages of Ag as compared to other coating material?
4- Describe the correlation of thickness to the properties of the coating material.
Comments on the Quality of English LanguageAcceptable
Author Response
Please see the attachment.

Reviewer 3 Report (New Reviewer)
Comments and Suggestions for Authors
The paper presents interesting results. However, there is little originality in the results presented. SERS sensing with femtosecond laser processing has already been demonstrated. HF wet etching was used previously to solve the problem of excessive roughness. The significance of the current results needs to be made more evident.
Most of my comments in the attached file are simple grammar corrections. But please pay close attention to content questions.

Comments on the Quality of English LanguageThere are a number of simple grammar corrections to be made.
Round 2
Reviewer 1 Report (Previous Reviewer 1)
Comments and Suggestions for Authors
I strongly recommend that this manuscript should be rejected.
The authors claim that they fabricated the sub-microstructures using a femtosecond laser, but fail to provide any evidence. They do not present their final SERS substrate with nanostructures. In the response letter, they repeat the same unsubstantiated claims. It would be better to remove the SERS content, including Figure 7, and just mention contact angle and transmittance, which can be discussed in another topic.
Comments on the Quality of English LanguageThe quality of the manuscript should be considered before discussing the English.
Author Response
Please see the attachment.

Reviewer 3 Report (New Reviewer)
Comments and Suggestions for Authors
My questions have been adequately answered in most cases. The fact that the beam size was calculated and not measured should be included in the manuscript.
Item 12 in the response: I would like to see the response [The reason for the higher transmittance than the plane in Fig. 6B is: According to the effective medium theory (EMT), the parameters of the anti-reflective sub-wavelength structure are determined by the following... ] added to the text in some form.
Author Response
Please see the attachment.

This manuscript is a resubmission of an earlier submission. The following is a list of the peer review reports and author responses from that submission.
Round 1
Reviewer 1 Report
Comments and Suggestions for Authors
The authors made patterns on a sapphire substrate using a laser and deposited silver to use as a SERS substrate. Before reviewing the significance of this manuscript, there are too many formal errors. It doesn't effectively present what the authors are trying to claim, and I question whether the authors would be willing to publish in this journal. This manuscript should be rejected and the authors need to revise the entire manuscript and resubmit. In addition, the following should be addressed. The authors have fabricated SERS substrates, but it lacks rationale and explanation for its design and should be rejected on contents as well.
1. The authors need to fix formal errors throughout the manuscript. There are many errors in engineering language, such as using u instead of micro, using the wrong units and spacing errors. These simple but critical mistakes make we question whether authors are serious about publishing in this journal. The abstract even includes figure number.
2. The authors adjusted the focus height by 3 micrometers. 3 micrometers is a very small height, and they should explain what equipment or parts they used to adjust it.
3. The authors describe microbusts forming inside the substrate in Figure 2 b,c. They should provide evidence to support this.
4. The authors use the SEM images in Figure 4 and the contact angles in Figure 5 to evaluate the surface of the substrate. However, these results describe the micro-scale and are not sufficient to describe the nanoscale at which SERS operates.
5. Are these samples evaluated after the Ag has been coated? Surface roughness should be discussed after the silver has been deposited.
6. Is there a reason why the nm units were used in Figure 6a and the um units in 6b?
7. Why is a discontinuity observed near 870 nm in Figure 6a?
8. The authors need to explain the calculation of the EF in more detail. We suggest including the calculations with numbers in the supplementary material.
9. The substrate is not uniform, how was the EF derived? Does a specific EFvalue meaningful on non-uniform substrate?
Comments on the Quality of English LanguageThe English in this paper needs a lot of improvement. It conveys meaning, but it's not good enough for engineering explanations.
Reviewer 2 Report
Comments and Suggestions for Authors
Dear Authors,
The article submitted for review contains important information in the area of innovative laser technologies. The procedure for obtaining micro-holes and surface modification of sapphire samples as well as the laser technological parameters used are described in detail. I think a few minor corrections should be made to improve the quality of the article.
1. A must!, more paragraphs (indents) should be introduced in Chapter 1 Introduction. Currently, the text "blends" and is difficult to read. It's similar in the remaining chapters - more paragraphs.
2. Please write the full name of the SERS abbreviation that was first used in the article in the abstract.
3. The caption under Figure 1 probably misspelled "Results and Discussion."
4. Figures 1A and 1B should include larger and better quality (more readable) photographs of the surface.
5. I suggest enlarging drawing 2A and drawing 6, especially the font. When printed on a piece of paper, you can't see much.
6. It is customary in scientific articles to provide the full name of the devices and the name of the manufacturer of the equipment with which experiments and measurements were performed. I suggest supplementing this information in this article.
Kind Regards
Reviewer
Reviewer 3 Report
Comments and Suggestions for Authors
OK, the work describes "micro-nano craters" in sapphire surfaces that form due to focused fs laser pulses. Ultimately, the gridded sapphire surface is coated with silver to show a SERS effect. The substance used for SERS is R6G, an optically highly active molecule, e.g., used for dye lasers. It is difficult not to see optical activity even for very small concentrations of R6G.
For a number of reasons this manuscript is not strong:
The term “micro-nano” is quite annoying, as if you were saying something like “g-kg material samples”. Rather nonsense. These here are sub-micro patterns.
In the manuscript often u instead of µ.
The abbreviation SERS is not explained here at all. As a not well-informed expert, I could also read here: “Super-Efficient Rhodamine Spectroscopy”. This is not possible for a journal contribution; the submitters and also the editors have to intervene before the review! (Surface Enhanced Raman Spect.)
Fig. 6. Where does the jump at 850 nm come from? That probably doesn't mean anything, but you can't just leave it out either. Is there actually anything that can be said from Fig. 6 other than that the material becomes cloudy? I suspect that there is no statement at all other than a bit of clouding. Why is the transmission in Fig. 6b even greater for wet etching than for flat? So isn't flat clean? What is with HF etching of flat substrates?
Fig. 7. After silver coating, an increased SERS signal from R6G occurs. However, this shows the complete loss of control in this work: For every scientific statement, comparisons and counter-tests must be made! So what is the SERS signal for silver on flat sapphire? What is the sub-microstructure of silver (depending on the coating process!) on different substrates? Do you actually need fs-laser-treated Al2O3 for SERS on silver? I don't believe. You could use any ceramic or glass plane or even metal surface. In my opinion, there is no clear connection at all between the elaborately created crater landscape and the R6G SERS on silver. Any randomly rough surface of the silver would be just as good. So what's the point?
All the work is pretty random. Of course, some µ or sub-µ craters appear in the Al2O3 after focused fs laser treatment. But it cannot be deduced here that these craters have any interesting or special properties, except roughness. Somehow nano or micro-roughness can be achieved differently, cheaper and in a better controlled manner, even on Al2O3
Also characteristic here: no line number for the review process, but this must also be clarified in advance by the editors!